# Lung Microbiota in Idiopathic Pulmonary Fibrosis, Hypersensitivity Pneumonitis, and Unclassified Interstitial Lung Diseases: A Preliminary Pilot Study

**DOI:** 10.3390/diagnostics13193157

**Published:** 2023-10-09

**Authors:** Milena Adina Man, Rodica Ana Ungur, Nicoleta Stefania Motoc, Laura Ancuta Pop, Ioana Berindan-Neagoe, Victoria Maria Ruta

**Affiliations:** 1Department of Medical Sciences-Pulmonology, Faculty of Medicine, University of Medicine and Pharmacy, 8 Victor Babeș Street, 400012 Cluj-Napoca, Romania; manmilena50@yahoo.com; 2“Leon Daniello” Clinical Hospital of Pneumophtysiology, 400371 Cluj-Napoca, Romania; victoria.suteu@yahoo.com; 3Department of Medical Specialties-Rehabilitation Medicine, Faculty of Medicine, “Iuliu Hațieganu” University of Medicine and Pharmacy, 8 Victor Babeș Street, 400012 Cluj-Napoca, Romania; ungurmed@yahoo.com; 4Research Center for Functional Genomics, Biomedicine, and Translational Medicine, “Iuliu Hatieganu” University of Medicine and Pharmacy, 8 Victor Babeș Street, 400012 Cluj-Napoca, Romania; laura.ancuta.pop@gmail.com (L.A.P.); ioananeagoe29@gmail.com (I.B.-N.)

**Keywords:** interstitial lung disease, lung microbiome, lung microbiota, idiopathic pulmonary fibrosis, hypersensitivity pneumonitis

## Abstract

(1) Introduction: Although historically, the lung has been considered a sterile organ, recent studies through 16S rRNA gene sequencing have identified a substantial number of microorganisms. The human microbiome has been considered an “essential organ,” carrying about 150 times more information (genes) than are found in the entire human genome. The purpose of the present study is to characterize and compare the microbiome in three different interstitial lung diseases: idiopathic pulmonary fibrosis (IPF), hypersensitivity pneumonitis, and nondifferential interstitial lung disease. (2) Material and methods: This was a prospective cohort study where the DNA of 28 patients with ILD was extracted from the lavage and then processed using the standard technique of 16S RNA gene sequencing. In a tertiary teaching hospital in the northern, western part of Romania, samples were collected through bronchoscopy and then processed. (3) Results: The same four species were found in all the patients but in different quantities and compositions: *Firmicutes*, *Actinobacteria*, *Proteobacteria* and *Bacteroides*. *Streptococcus* was the most prevalent genus, followed by *Staphylococcus* and *Prevotella*. Statistically significant differences in the OUT count for the ten most abundant taxa were found for the genus: *Gemella*, *Actinobacteria*, *Prevotella*, *Neisseria, Haemophilus*, and *Bifidobacterium*. The comparative analysis showed a richer microbiota in patients with IPF, as shown by the alpha diversity index. (4) Conclusions: In interstitial lung diseases, the microorganisms normally found in the lung are reduced to a restricted flora dominated by the *Firmicutes* family. These changes significantly disrupt the continuity of the observed bacterial pattern from the oropharynx to the bronchial tree and lung, possibly impacting the evolution and severity of interstitial lung diseases.

## 1. Introduction

Although historically, the lung has been considered sterile, culture-independent technologies based on 16S rRNA gene sequencing have identified an important number of microbes [1]. The human microbiota, whose composition has not been fully identified, is comprised of bacteria, fungi, and viruses [2]. Colonizing the human body with bacteria and microorganisms in symbiosis with the host, creates a community defined as the “microbiome”. The microbiome refers to the collective genomes of microorganisms in a specific environment, while the microbiota is the community of microorganisms found in those specific environments [3]. The lung microbiome has been dynamic over time. The maintenance of local homeostasis depends on bacterial migration (through microaspiration), elimination (cough, mucociliary clearance), and dynamic reproduction (through the activity of the immune system), mechanisms that are influenced by the local environment and the general health status of the host [4,5]. The external microbial exposure—the microorganisms from the patient own environment, the microbe–microbe interaction, and the host–microbe interaction constitute the lung microbiome of the adult [6]. The interactions between members of the microbiota can be positive (commensalism, synergism, mutualism) or negative (antagonism, parasitism, competition), with more than 100 trillion symbiotic microorganisms living on and inside the body and playing an important role in maintaining health. Trillions of microbes have evolved alongside humans and now live on and within them, influencing human health and disease. The human microbiome has been considered an “essential organ”, carrying about 150 times more genes than the entire human genome [7]. From the beginning until the end of our life, the airway microbiota varies and correlates with age [8]. With the whole physiological evolution, the microbiota characteristic of the lung has a common trait; the commensal species must possess defence mechanisms against alveolar macrophages, with impressive phagocytic capacities [6]. The bacterial families identified in the lungs of healthy individuals include *Proteobacteria*, *Firmicutes*, and *Bacteroidetes*, while from a genus perspective, the most common are *Streptococcus, Prevotella* and *Veillonella.* The operational taxonomic unit (OUT) defines clusters of similar 16S rRNA gene sequences. Each OTU represents a taxonomic unit of a bacteria family or genus depending on the sequence similarity threshold [1]. Using samples from the lower airways through bronchoalveolar lavage (BAL), lungs are considered to have a low bacterial load compared to the upper airways examined using sputum samples in healthy individuals [9]. Metagenomic studies offered the possibility to reconstitute the viral genome, with the emergence of the virome, embedded in the human ecosystem of commensal microorganisms but studied as a separate entity [2]. Technologies using 16S rRNA gene sequencing have significantly revolutionized our knowledge in the field, proving that lungs shelter a wide range of microbial species that are important in maintaining health [10]. A similar technique is used in case of viral and fungal identification, but with the detection of 18S rRNA in the case of fungi and with nucleic acid sequencing and PCR in the case of viruses [11]. Sequencing targeted regions in the ribosomal locus, such as the 18S rRNA gene or internal transcribed spacer (ITS) are also used in fungi identification. In the case of viruses, however, shotgun metagenomics—which sequences the entire nucleic acid extracted from the sample—is still primarily used to investigate them because viruses lack conserved nucleic acid sequences [12]. The diffuse interstitial diseases, a field still insufficiently exploited, represent an interest to researchers due to the wide range of possibilities for learning and discovering new markers, which would facilitate the diagnosis. On the other hand, the lung microbiome has yet to be researched in Romania, and more data are needed to characterize it accurately.

**Aim of the study:** Therefore, the purpose of the present study was to characterize and compare lung microbiomes, in terms of bacterial load, composition, and diversity, in three different interstitial lung diseases: idiopathic pulmonary fibrosis (IPF), hypersensitivity pneumonitis (HP), and undifferentiable interstitial lung (ILD) using the standard technique of 16S RNA gene sequencing. The study’s second objective was to perform a comparative analysis of the microbiota of patients suffering from these different diseases.

## 2. Materials and Methods

**Study design:** This was a prospective cohort study carried out in “Leon Daniello” Pneumophysiology Hospital Cluj-Napoca between 2019 and 2022, on a group of 28 patients (10 IPF, 10 HP, and 8 with unclassifiable ILD). The hypothesis we started from was that the lung has a unique imprint in various conditions, and its identification can help us better understand a disease and its evolution. The study population included all patients over 18 years old with confirmed interstitial lung disease that met all the inclusion criteria: confirmed interstitial lung disease, as per ATS/ERS/JRS/ALAT 2018 criteria, and the first diagnosis of ILD. The exclusion criteria were: a history of neoplasia, regardless of location; association of known chronic obstructive pulmonary diseases (asthma, COPD); known genetic disease (cystic fibrosis), diagnosis of pneumoconiosis (silicosis), or a history of pulmonary tuberculosis; confirmation of a new/onset bronchial obstructive syndrome after pulmonary respiratory testing; acute infectious disease (symptoms suggestive of fever; biologically suggestive signs: leukocytosis ≥ 12,000/mm^3^, procalcitonin ≥ 0.5 ng/mL; sputum examination positive for various germs); patients who received antibiotic treatment 14 days before admission, regardless of cause; severe respiratory failure requiring NIV; severe cardiovascular disease (cardiac arrhythmias, unstable angina pectoris, recent myocardial infarction ≤ 6 weeks, pulmonary thromboembolism ≤ 3 months, coagulation disorders, thrombocythemia ≤ 50,000/mL); and documented psychiatric disease or non-compliant patients or patients with low comprehension. The enrolled patients met all the inclusion criteria and none of the exclusion criteria. After diagnosis, patients were divided into three groups: patients with idiopathic pulmonary fibrosis (IPF), hypersensitivity pneumonitis (HP), and unclassifiable interstitial lung disease (unclassified ILD). Written informed consent was obtained from all participants. All participants consented to a bronchoscopy examination at our hospital for diagnostic purposes, and the results obtained were discussed with the multidisciplinary team weekly. The final IPF, HP, and unclassifiable ILD diagnosis was established in multidisciplinary discussion according to ATS⁄ERS⁄JRS⁄ALAT 2018. Medical history was collected from all participants. A set of routine pre-procedure tests were carried out including physical examination, electrocardiogram, pulmonary function testing, computed tomography, routine blood count, and blood coagulation function analysis. Due to the need to use invasive procedures and the ethics committee’s reluctance, there was no comparatively healthy group. That is why the cohort has only patients with ILD and no control group. The study methodologies conformed to the standards set by the Declaration of Helsinki and were approved by the ethics committee of the “Iuliu Hațieganu” University of Medicine and Pharmacy (no. 345/1.10.2019). For sample preparation, we used the BAL collection method. The fibro bronchoscopy procedure was performed according to the procedural standards and recommendations. From the total BAL, the same laboratory physician separated in the first 15 min 2 × 2 mL of pathological fluid, which was stored at −80 °C degrees in the refrigerator of the hospital laboratory. The samples were transported on ice, all at once, to the Research Centre for Functional Genomics, Biomedicine, and Translational Medicine, within the UMF Cluj to process the lung microbiome. The remaining sample was kept for further research (approximate 1 mL). For bacterial DNA extraction, from the lung lavage of the selected patients, we took 1 ml for bacterial DNA extraction using the QiAmp DNA Microbiome kit (Qiagen, Hilden, Germany) using the manufacturer’s protocol. The obtained DNA was quantified using a NanoDrop spectrophotometer (ThermoScientific, Waltham, MA, USA), and we obtained concentrations between 0.2 and 4.6 ng/µL. For metagenomic analysis, we used next-generation sequencing technology. In total, 12 µL of each DNA sample was used to amplify the 16S bacterial hypervariable regions using the Ion 16S Metagenomics kit (ThermoFisher Scientific). After amplification, the samples were purified using the AmPure XP beads (BeckmanCoulter, Brea, CA, USA) and quantified quantitatively and qualitatively using the Agilent Bioanalyzer and the High Sensitivity DNA kit (Agilent, Santa Clara, CA, USA). Further, 100 ng of the purified sample was used for sequencing library synthesis using the Ion Plus Fragment Library kit (ThermoFisher Scientific). During library synthesis, barcodes (Ion Xpress Barcode Adapters 1–16 kit (ThermoFisher Scientific) were added to each sample to sequence seven libraries in a run. The sequencing libraries were quantified using the Ion Universal Library Quantification kit (ThermoFisher Scientific) and the real-time PCR Viia 7 instrument (ThermoFisher Scientific). Each library was diluted to 10 pM, and then 3 µL from each of the 7 libraries were mixed and used for template synthesis using the Ion PGM HiQ View OT2 kit (ThermoFisher Scientific) and Ion OneTouch 2 instrument (ThermoFisher Scientific). The obtained template was loaded on an Ion 318 chip (ThermoFisher Scientific) and sequenced on the Ion Torrent Personal Genome Machine (ThermoFisher Scientific) with the help of the Ion PGM Hi-Q View Sequencing kit (ThermoFisher Scientific). The obtained template was loaded on an Ion 318 chip (ThermoFisher Scientific) and sequenced on the Ion Torrent Personal Genome Machine (ThermoFisher Scientific) with the help of the Ion PGM Hi-Q View Sequencing kit (ThermoFisher Scientific) and in single read mode. The output of the sequencing runs showed a mean reads length of 225 bp, with 277,470 to 1,113,955 reads per sample, Q20 reads between 4,867,974 and 231,062,125 bases, and an output of up to 1.3 Gb of data.

**Statistical Analysis:** We conducted statistical analysis in GraphPad Prism v9. Alpha diversity metrics were used. These indices show the composition of an ecological community concerning its richness (number of taxonomic groups), evenness (distribution of abundances of the groups), or both. The ones we used in our study were observed operational taxonomic units (observed OTUs), Chao 1 to non-parametrically estimate the richness of the species [13], the Shannon index to estimate the richness and evenness of species present in a sample considering the distribution of strains belonging to each species 14, and the Inverse Simpson index to measure the probability that two randomly selected objects in a sample belong to the same species [14]. Principal coordinates analysis (PCoA) was done using the EMPEROR software (version 5.20.2.0) from the Ion Reporter workflow.

## 3. Results

A total of 28 patients with interstitial lung disease (10 with IPF, 10 with HP, and 8 unclassifiable ILD) underwent metagenomic analysis. Patients’ demographic data are shown in Table 1. Most of the patients included in the study were females, non-smokers, with a mean age of 60 (see Table 1). Although alveolar macrophages predominated in BAL in all groups, mixed alveolitis with predominant lymphocytosis was observed in subjects with HP 0.6 (0.19–0.93). Radiologically, honeycomb-like changes were found in 50% of IPF patients, and 60% of patients with HP had ground glass predominance on HRCT. 

The bacteriological analysis from the brochoalveolar lavage were negative–negative cultures, including those for Mycobacterium tuberculosis. 

Following the secondary analysis with the Ion Reporter v5.18 software, we obtained information about the microbial profile of each patient. Every sample was separately analyzed in terms of the identified bacterial community. Each case analyzed comparatively is shown in Figure 1A–C. 

The HP group had a predominance of *Firmicutes*, followed by Actinotinobacteria, *Proteobacteria*, and the last ones being Bacteriotides and Fusobacteria (see Figure 1A) 

The IPF group presented a more affluent bacterial community than HP or unclassifiable ILD, both at the family and genus levels. IPF patients had a low percentage of *Firmicutes* and a high percentage of *Actinobacteria*, *Fusobacteria*, and *Bacteroides* (see Figure 1B)

The unclassifiable ILD group microbiome was dominated by *Actinobacteria*, followed by *Firmicutes*, *Bacteroides*, *Fusobacteria*, and then *Proteobacteria* (see Figure 1C).

Considering all three groups, we observed a similar microbial composition dominated by: *Firmicutes*, *Actinobacteria*, *Proteobacteria*, *Bacteroidetes*, and a low average of *Fusobacteria* in all samples, except for unclassifiable ILD, where the average of identified *Fusobacteria* is higher. 

The IPF microbiota was dominated by *Firmicutes* (49%), *Actinobacteria* (29%), *Proteobacteria* (15%), and *Bacteroides* (3.7%). The same species dominated in HP with a slight excess of *Firmicutes* (51%), *Actinobacteria* (30%), and *Bacteroides* (4.5%) and a reduction of *Proteobacteria* (12%). *Firmicutes* dominated in the IPF and HP groups without a statistical correlation being found. Compared to HP and IPF, unclassifiable ILD showed significantly higher *Bacteroides* (6.3% in ILD) (Figure 2).

We identified a different number of microorganisms in each of the three tested groups of patients. At the genus level, IPF patients had the highest number of different pathogens (394), followed by unclassifiable ILD (306) and the least HP (304). All the alpha diversity indices evaluated in our study (observed OTUs, Chao 1, Shannon, and inverse Simpson) differed in the evaluated groups (IPF, HP, and unclassifiable ILD), as seen in Figure 3A–D. The IPF group had higher values when compared with patients with HP or patients with the unclassifiable ILD group. The latter had the lowest levels in observed OTUs and Chao 1 category and higher levels in Shannon and inversed Simpson indexes. The PCoA plots done on all the samples together or separately present heterogeneity in the microbial distribution of each of the three groups, as seen in Figure 4. We can also observe that each group has its outlier, and in the PCoA plot of all the samples together, we can separate two groups composed of samples from all three categories.

When analyzing the ten most abundant taxa at the genus level in all three groups of patients, we observed that *Streptococcus* was the most prevalent genus, followed by *Staphylococcus* and *Prevotella* (Figure 5). 

Statistically significant differences in the OUT count for the ten most abundant taxa were found for the genus: *Gemella*, *Actinobacteria*, *Prevotella*, *Neisseria*, *Haemophilus*, and *Bifidobacterium*. The *Gemella* and *Actinobacter* genus presented lower OTU counts in HP and unclassifiable ILD than IPF. The *Prevotella* genus was lower in HP compared to IPF, while *Neisseria* and *Haemophilus* were lower in undifferentiated ILD compared to IPF. *Bifidobacterium* had higher OTU counts in IDH endif than in IPF (Figure 6).

## 4. Discussion

The current pilot study evaluated microbial lung profiles in patients with IPF, HP, and undifferentiated ILD disease using the standard 16S RNA gene sequencing technique. The IPF microbiota was dominated by *Firmicutes* (49%), *Actinobacteria* (29%), *Proteobacteria* (15%), and *Bacteroides* (3.7%). In patients with HP, we found the same families with a slight excess of *Firmicutes* (51%), *Actinobacteria* (30%), and *Bacteroides* (4.5%) and a reduction of *Proteobacteria* (12%). The unclassifiable ILD group showed significantly higher *Bacteroides* (6.3% in ILD). *Streptococcus* was the most prevalent genus, followed by *Staphylococcus* and *Prevotella*. Statistically significant differences in the OUT count for the ten most abundant taxa were found for the genus: *Gemella, Actinobacteria, Prevotella, Neisseria, Haemophilus,* and *Bifidobacterium*. The comparative analysis showed a richer microbiota in patients with IPF, as shown by the alpha diversity index. The lung microbiota significantly affects disease progression and survival in various pulmonary diseases [15]. Recently, many studies have used 16S rRNA gene pyrosequencing technology to explore lung microbiota in healthy persons and interstitial disease [16]. Dysbiosis of the lung microbiome could influence the occurrence and evolution of ILD [17].

The increased prevalence of *Firmicutes* and *Proteobacteria* is the prerogative of chronic diseases [18], and this change in commensal bacterial proportions determines the reduction of diversity and consequently the appearance of dysbiosis, with a stipulated role in fibrosing ILD [19]. Existing data on the variety of lung microbiota in various pathologies has proposed a microbial print” associated with diagnosis, illness severity, and prognosis [13]. The pathophysiological mechanisms triggering fibrosis are mainly unknown. Decreased microbial diversity has been associated with the possible evolutionary character of ILD. The lung microbiota has been proposed as a mediator of inflammation [14]. The bacterial families that dominated BAL in patients with ILD (HP, IPF, and unclassifiable ILD) in the present study were *Bacteroidetes*, *Firmicutes*, *Proteobacteria* and *Actinobacteria*. Molyneaux et al. aimed to find the role of the lung microbiota in IPF pathogenesis and progress and found a two-fold increase in bacterial burden in the lung microbiota of IPF patients compared with healthy controls and COPD [20,21]. The existing pathogens (viruses, bacteria, and environmental fungi) have been, for a long time now, hypothesized to play a role in the pathogenesis of ILDs [16]. New data revealed that the human distal airways harbor several bacterial species. These microorganisms form a unique ecological community and changes in their composition or density are associated with disease progression, acute exacerbations, and mortality in idiopathic pulmonary fibrosis [16]. The *Firmicutes* family is associated with an increased risk of IPF progression, and its dominant genera, *Streptococcus* and *Staphylococcus*, are correlated with reduced survival time [22]. IPF is not simply the direct result of architectural distortion and parenchymal destruction; microbiota change could influence the immune system, systemic and local inflammation, and, consequently, the progression and maintenance of pulmonary fibrosis. Discussions between different microbiota of the same or different organs could underlie the understanding of local anarchic dysfunctions [16,17]. Microbial–host interactions are essential in the pathogenesis of pulmonary fibrosis, as is suggested by the lung dysbiosis and peripheral blood mononuclear cell transcriptomic expression of immune pathways [13]. Most studies of microbial imbalances in the upper and lower airways have used sputum samples, which may be affected by oropharyngeal contamination. Bronchoscopy-based studies minimize upper airway contamination and provide more accurate microbiota data. However, due to the invasive techniques required to collect BAL, they generally involve a small number of cases [16] which is a secondary factor that should not be omitted in the analysis of the microbiota because, in the context of ILD, most patients are old (60 years old), especially those with IPF. This finding cannot be attributed to the casuistry of the current study because, in IPF, the mean age was 53. Thus, we cannot correlate the reduced diversity with the age-related factor [15]. In the COMET study [20], 55 BAL samples were analyzed in order to evaluate the progression of interstitial lung diseases. The most identified bacteria were *Prevotella*, *Veillonella,* and *Escherichia* spp., *Streptococcus* spp., or *Staphylococcus* spp. at baseline, which were associated with the evolution of interstitial disease. Even more, the load of bacteria at the time of diagnosis was associated with rapid and a high risk of mortality (HR 4.59). A small microbial diversity was also observed in the IPF cohort with an abundance of *Veillonella*, *Neisseria*, *Streptococcus,* and *Haemophilus* spp. Recently, Yin et al., for the first time, analyzed the virome in patients with stable IPF by using next-generation RNA sequencing. Analyzing lung tissue samples from 28 patients with IPF and 20 controls who underwent surgical lung biopsy, a sporadic presence of viral RNA in tissue specimens was detected by real-time quantitative PCR. Observing the abundance of *Veillonella*, *Neisseria*, *Streptococcus,* and *Haemophilus* spp., there was no difference between the lungs of IPF patients and controls [1]. In 2020, Invernizzi et al. [20] assessed the role of microbiota in patients with HP using IPF patients as diseased controls. Their aim was to notice a pattern of the microbiome in different lung diseases. A total of 110 patients with chronic hypersensitivity pneumonitis (CHP) were retrospectively recruited. Their BAL microbiota (sequenced with PCR amplification of the 16S rRNA genes) was compared with 45 IPF patients and 28 control subjects. RNA 1. To understand the respiratory microbiome–immune system interactions in health and disease [23], it is crucial to understand the mechanism that might trigger or perpetuate a condition. There are limits in doing that because most bacteria can only be identified by genus or family. Shotgun metagenomics gene sequencing can capture valuable information about a microbial community (bacterial, fungal, or viral). Neither shotgun metagenomics nor 16S rRNA gene sequencing can distinguish between live and dead bacteria [12]. In the lung microbiome, the dominant phyla are *Firmicutes* and *Bacteroidetes*, and the main bacterial genera are *Prevotella*, *Porobacteria*, and *Streptococcus*. It has been proved that the lung is not a sterile organ and it comprises a complex and diverse bacterial community [12]. The understanding of normal lung and the microbiome of different lung diseases could ease the way for long-term antibiotics in these patients. Different studies showed that the use of prophylactic azithromycin or doxycycline were beneficial for preventing ILD exacerbation. The use of three months of co-trimoxazole (trimethoprim–sulfamethoxazole) proved to be beneficial for the life quality in patients with fibrotic ILDs. This hypothesis was sustained by other studies that prove that oral co-trimoxazole for 12 months, alongside the usual treatment in patients with fibrotic idiopathic interstitial pneumonia, increased life quality. However, it could be argued that the reduced mortality in the antibiotic group might be attributed to a decrease in the rate of respiratory infections, given that most patients on “usual treatment” were taking immunosuppressants. Recently, other studies have discussed the role of long-term antibiotics on relevant outcomes IPF patients. There is a disparity between the studies as other studies do not support long-term antibiotic administration as there were no major improvements in cough frequency and neither in life quality. Concerning adverse effects, diarrhea was more frequent in patients treated with azithromycin than placebo (43% vs. 5%; *p* = 0.03) RNA [1]. While microbiome-based therapies are intriguing, restoring the ecological niche in the lung is essential. A diseased lung forms a different habitat for microbes, favoring a diverse microbial flora. In cystic fibrosis and bronchiectasis, *Haemophilus* spp. and *Pseudomonas* spp. increasingly dominate the lung microbiota, while specific antibiotic treatment promotes their outgrowth. This could result from reduced competition from commensals destroyed by antibiotics or cystic fibrosis lungs’ persistent aberrant ecological niche. Supporting the latter, a recent study showed that using ivacaftor to increase CFTR function in patients with the G511D mutation, thereby partially restoring mucociliary clearance, rapidly decreasing the *P. aeruginosa* burden in the sputum of patients. However, *P. aeruginosa* was not eradicated, and outgrowth of *P. aeruginosa* was observed after one year of treatment, possibly due to incomplete restoration of the niche [24]. Multiplex PCR assays do not accurately distinguish DNA from viable versus non-viable organisms. In this case, initial detection does not necessarily mean that the patient has to be treated or, if he is already being treated, there is a need to prolong antimicrobial therapy. Even more culture-independent methods applied to samples with low microbial biomass are far more predisposed to sequencing noise and DNA contamination. In this matter, microbial RNA metatranscriptome sequencing methods may be beneficial as detection of microbial RNA generally suggests the presence of viable microbes, given that RNA is rapidly degraded after cell lysis. There are more and more data that support the association between lower airway bacterial burden and disease progression-free survival among patients with IPF. Natalini et al. reported a correlation between alveolar inflammatory and fibrotic cytokines and lung microbiota diversity and composition. Lower Shannon diversity indices (within-sample diversity) were associated with higher concentrations of IL-1Ra, IL-1β, CXCL8, MIP1α, G-CSF, and EGF [13]. In addition, alveolar concentrations of IL-6 were positively correlated with the relative abundance of the lung *Firmicutes* phylum, whereas alveolar IL-12p70 was negatively correlated with the relative abundance of the lung *Proteobacteria* phylum.

The major limitation of human studies is the inability to determine the directionality of observed associations. Animal studies are more eloquent: in a murine bleomycin model of pulmonary fibrosis, microbial dysbiosis appeared before lung injury and persisted until the development of fibrosis. Germ-free mice exposed to bleomycin-induced pulmonary fibrosis had a mortality benefit compared to conventional mice. In another murine model, *Bacteroides* and *Prevotella* species were linked with increased fibrotic pathogenesis through IL-17R signalling. All this information advocates that specific microbial exposures may act as persistent stimuli for repetitive alveolar injury, contributing to pulmonary fibrosis and inflammatory injury in this disease.

However, a lack of consistent findings across studies makes it difficult to define lung dysbiosis clearly. It is possible that even within a single disease process, there are many forms of dysbiosis, a concept that has previously been referred to as the ‘Anna Karenina principle’ about Tolstoy’s writings that “happy families are all alike; every unhappy family is unhappy in its way” [13]. The lung microbiome differs between patients with DM and RA associated with ILD [25]. In the present study, Streptococcus was high in subjects with IPF compared to HP and unclassifiable ILD, and Staphylococcus was found in relatively equal proportions in the three cohorts. The current research may support the theory that changes in the microbiome characterized by an increased abundance of bacteria belonging to the *Firmicutes* family may play a role in the pathogenesis of ILD [1,16].

Bacterial commensalism favors the mutual survival of specific genera, increasing each other’s potency. Identifying the genus *Streptococcus* in IPF may explain the anarchic lesional expansion because it produces pneumolysin, a toxin that stimulates fibrotic progression by damaging the alveolar epithelium. *Veillonella* spp. (of the same *Firmicutes* family) have been shown to induce the growth of *Streptococcal* biofilm, frequently found together [26].

*Veillonella* is a commensal, anaerobic, Gram-negative coccus typically found in the oral cavity and intestine, with identification in the lower respiratory tract in severely immunocompromised hosts (HIV, liver cirrhosis, excessive alcohol consumption) [27]. The association found in the current study, *Streptococcus* and *Veillonella*, in IPF cases, from the moment of diagnosis, may support the hypothesis of fibrotic progression and extensive lung lesions from the early stages of the disease.

Increased bacterial load in BAL at the time of diagnosis of IPF and other ILDs predicts a rapid, progressive course with an increased risk of death [16].

Compared to conventional bacterial cultures, sequencing methods have two advantages: identifying unusual pathogenic bacteria that are not detectable by conventional microbiological methods and identifying the pathogenic role of “commensal” bacteria [28]. The depletion of the normal microbiota with “overgrowth” of a single bacterial species may indicate that this is clinically relevant, for example, the growth of *Firmicutes*. This is essential because a clinician usually distinguishes between harmless commensals, “facultatively pathogenic”, and “pathogenic” bacteria. It is essential to emphasize that traditional cultures and molecular methods should be viewed as complementary [29]. Although the respiratory system has a surface area greater than 70, m^2^—which—is the size of a tennis court. It is in direct contact with the environment; the concept above pervaded knowledge of the respiratory system until the early 21st century, when the first studies based on molecular techniques for identifying bacterial DNA revealed the presence of genetic material from microorganisms in the lower respiratory tract.

The microbiome and its changes likely directly influence the natural history of respiratory diseases and a change in the microbiota resulting from antibiotic treatment of infectious respiratory tract diseases. In addition, increasing knowledge of the lung microbiome has brought about a discussion of a possible distinction between those bacterial species that are pathogens and those that behave as commensals in the composition of our physiological microbiome [11,30,31,32]. The main limitation of the present study is the relatively small sample size and the absence of control groups. Another thing that worth mentioning is the pathogenic nature of the discovered microorganism. As mentioned in the paper, all the included patients were stable, and none had positive culture in the lavage so can we say there are pathogens? What is the significance of their presence in the lung? Our preliminary findings require confirmation in more extensive studies and the evaluation of significance. Importantly, however, our study revealed the differences in lung microbiota among patients with various interstitial lung diseases, and it is the first in our country, although only in one center. We also knew that the greater the number of samples collected, the better the study’s statistical power. However, BALF sample collection took much work to implement. This study demonstrated that although the ILD microbiota consists of the same four families as healthy individuals, the identified bacterial load is higher. Significant differences in the composition and diversity of their microbiota may dictate the occurrence and evolution of lung diseases.

## 5. Conclusions

During interstitial lung diseases, the microbiota transforms into a restricted flora dominated by the *Firmicutes* family, which includes most potentially pathogenic microorganisms, parallel with a decline in *Bacteroides*. These changes significantly disrupt the continuity of the observed bacterial pattern from the oropharynx to the bronchial tree and lung, possibly impacting the evolution and severity of interstitial lung diseases. This prospective observational cohort study provides the first evidence of the microbial composition in Romania, providing details on the microbial footprint of patients with the most particular and frequent forms of ILD: IPF, HP, and unclassifiable ILD. Given their unpredictable evolution, knowing the microbiome at the time of diagnosis can provide information on evolutionary, prognostic, and therapeutic possibilities. Future research should address the correlation between the total bacterial load, abundance of microbial genera, and progression of pulmonary fibrosis.

These critical studies will lay the foundation for individualized, microbiome-based therapies, and eventually improve survival.

## Figures and Tables

**Figure 1 diagnostics-13-03157-f001:**
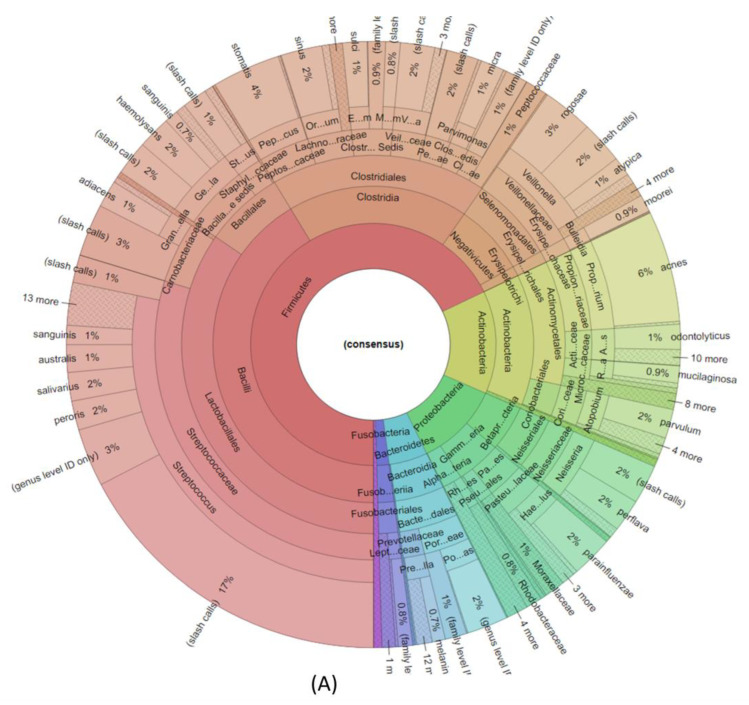
Krona diagram of the microbial profile of (**A**) a sample of HP, (**B**) IPF, and (**C**) a sample of unclassifiable ILD.

**Figure 2 diagnostics-13-03157-f002:**
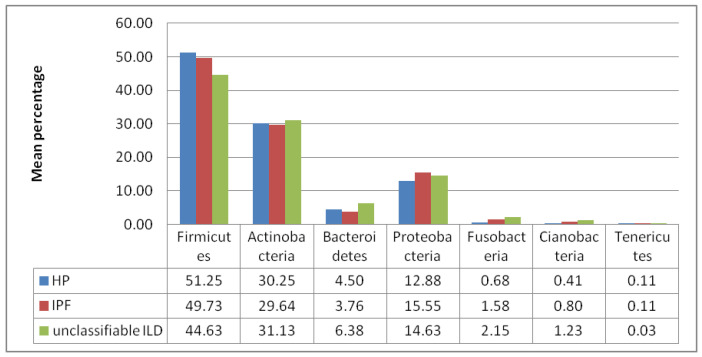
The average percentage of bacterial families found in the three groups.

**Figure 3 diagnostics-13-03157-f003:**
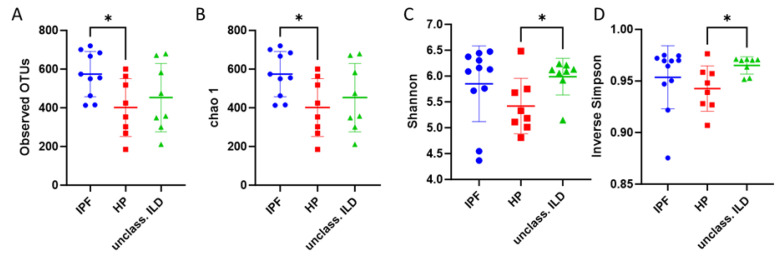
Alpha diversity plots for the study’s three patient groups: (**A**) Observed OTUs, (**B**) Chao 1; (**C**) Shannon index, and (**D**) inverse Simpson index. “*” represent the *p* value of the analysis, which in our case is <0.05, which means the different is statistical significant. Blue—patients with idiopathic pulmonary fibrosis; Red—patients with hypersensitivity pneumonitis; Green—patients with unclassified ILDs (interstitial lung diseases).

**Figure 4 diagnostics-13-03157-f004:**
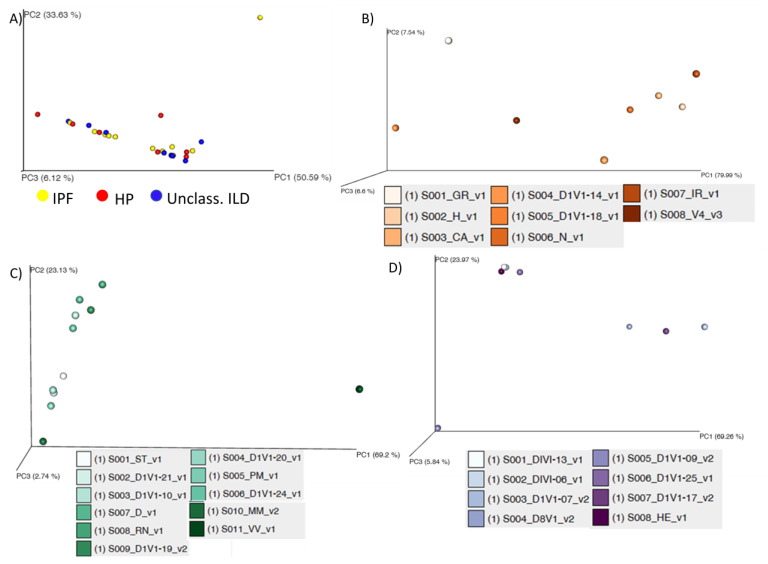
Euclidean PCoA plots: (**A**) all samples, (**B**) HP samples, (**C**) IPF samples, and (**D**) unclassifiable ILD. IPF—idiopathic pulmonary fibrosis; HP—hypersensitivity pneumonitis; Unclassified ILDs—unclassified interstitial lung disease.

**Figure 5 diagnostics-13-03157-f005:**
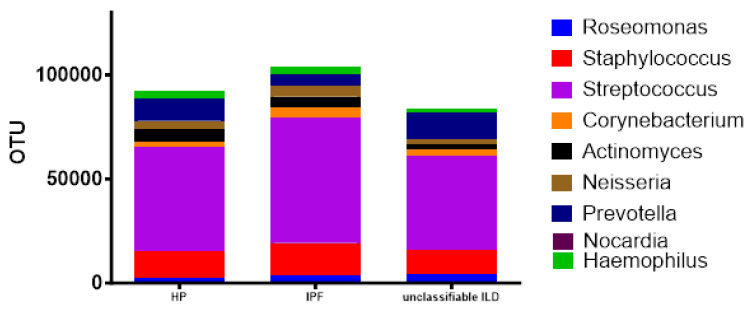
Most abundant taxa at the genus level for the three groups. IPF—idiopathic pulmonary fibrosis; HP—hypersensitivity pneumonitis; Unclassified ILDs—unclassified interstitial lung disease.

**Figure 6 diagnostics-13-03157-f006:**
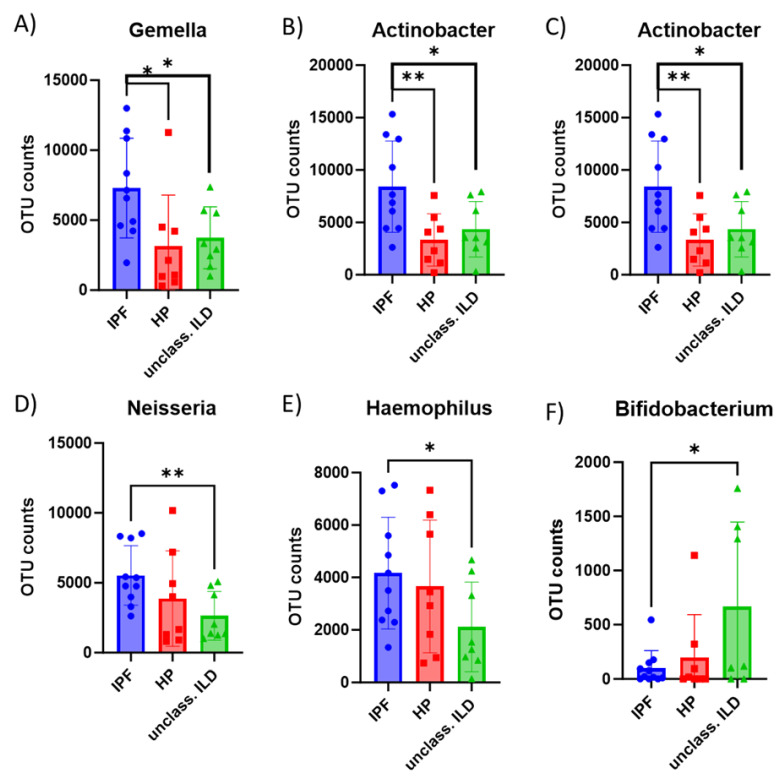
Comparison of OTU counts for IPF compared to HP and undifferentiated ILD (**A**) *Gemella*, (**B**) *Actinobacteria*, (**C**) *Prevotella*, (**D**) *Neisseria*, (**E**) *Haemophilus*, and (**F**) *Bifidobacterium*. “*”is the *p* value of the analysis and is < 0.05; “**”is the *p* value of the analysis and is < 0.005. Blue—patients with IPF—idiopathic pulmonary fibrosis; Red—patients with HP—hypersensitivity pneumonitis; Green—patients with Unclassified ILDs—unclassified interstitial lung disease.

**Table 1 diagnostics-13-03157-t001:** Demographic data of evaluated patients.

Parameter	IPF (*n* = 10)	HP (*n* = 10)	Unclassified ILDs (*n* = 8)	*p* Value
Age	53.3 ± 12.96	62.3 ± 5.79	58.75 ± 16.36	NS
Gender, F, *n* (%)	6 (60%)	5 (50%)	5 (62.5%)	NS
Active smoker	1 (10%)	3 (30%)	4 (50%)	NS
Never smoker	8 (80%)	7 (70%)	4 (50%)	NS
Former smokers	1 (10%)	0	0	NS
Blood neutrophils	5.58 ± 2.41	4.67 ± 2.42	3.97 ± 0.99	NS
Blood lymphocites	1.89 ± 0.58	1.76 ± 0.64	1.68 ± 0.9	NS
NLR	2.99 ± 1.15	3.04 ± 2.28	2.96 ± 1.39	NS
Honey combing HRCT	5 (50%)	4 (40%)	0 (0%)	<0.05
Macrophage (BAL)	1.26 (0.55–5.16)	0.56 (0.24–2.18)	1.06 (0.56–1.89)	NS
lymphocites (BAL)	3.78 (1.72–8.08)	5.7 (1.80–14.69)	8.35 (5.25–13.74)	NS
Neutrophils (BAL)	0.53 (0.11–1.23)	0.6 (0.19–0.93)	0.33 (0.03–1.68)	NS

IPF—idiopathic pulmonary fibrosis; HP—hypersensitivity pneumonitis; Unclassified ILDs—unclassified interstitial lung disease; NS—non significant clinically; NLR—neutrophils/lymphocytes ratio; HRCT—high-performance computer tomography; BAL—bronchioloalveolar lavage.

## Data Availability

Data are available, if requested.

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
