# Peer review of "Lung Microbiota in Idiopathic Pulmonary Fibrosis, Hypersensitivity Pneumonitis, and Unclassified Interstitial Lung Diseases: A Preliminary Pilot Study"

_diagnostics, 2023, doi:10.3390/diagnostics13193157_

Round 1

Reviewer 1 Report

Dear Author manuscript is well written. Although some of the queries needs explaination 

1- How will you differentiated pathogenic or non pathogenic organism in BAL fluid of different ILDs like IPF, HSP and other.

2- What are the diagnostic criterias to confirm different ILDs in your study.

3- What is the culture report of BAL Fluid in your study?

Some minor english correction is required.

Author Response

First and foremost, thank you very much for your interest in our work, for your time and support. Suggestions are always welcome as they can help us improve the paper. 

Dear Author, the manuscript is well written. Although some of the queries needs explanation.

1.How will you differentiated pathogenic or nonpathogenic organism in BAL fluid of different ILDs like IPF, HSP and other

As mentioned in our paper, there is data that show that the lung is not sterile, and disease onset is actually a form dysbiosis in the lung microbiota What we did is follow: we have looked for the bacteria that is find in the lung in healthy persons and saw that those species are also present in the IPF patients, but only in different quantities. We have purposely looked for bacteria that we knew from previous research that pertain to health lung microbiota. We do not know if we can call them pathogens or not yet, as we do not know the significance of their presence in the lung. All the enrolled patients were stable, without any signs of infections, and as the BAL was negative in all patients (we have loik for specific bacterial agents including Mycobacterium tuberculosis) we supposed that there is no pathogen. BAL was used only for HP differential. In the other ILD was used as a aditionla information. 

2- What are the diagnostic criteria to confirm different ILDs in your study.

  • The diagnosis was made after a multidisciplinary meeting and in the light of current guidelines.

3- What is the culture report of BAL Fluid in your study?

We took test from all the patients and all the cultures were negative.

Reviewer 2 Report

Review: Diagnostics 2023, 13, x. https://doi.org/10.3390/xxxxx

The publication by Man et al., titled "Lung Microbiota in Idiopathic Pulmonary Fibrosis, Hypersensitivity Pneumonitis, and Unclassified Interstitial Lung Diseases: A Preliminary Pilot Study," presents new data in the fields of human microbiota and microbiome. For a long time, the human lower respiratory tract was believed to be sterile, but modern techniques, such as metagenomic sequencing, have revealed a diverse community of microorganisms residing in this human body part.

The manuscript is composed of an abstract, followed by five main sections, and references.

Abstract:

The abstract is clear and well-organized. It effectively summarizes the key points of the manuscript.

Consider adding a brief summary of the key findings mentioned in the abstract. This will provide a more comprehensive overview for readers.

Some minor editorial corrections are needed in terms of formatting, such as removing or bolding numbers before section titles (e.g., lines 20, 25).

Lines 29-30: The same four species […]. Consider changing the word “species” to “taxa,” since Firmicutes, Actinobacteria, and Proteobacteria are phyla, and as Bacteroides and Streptococcus are species, the term "species" at the beginning may be confusing for some readers.

In conclusion (Line 35-36), consider changing the sentence: […] the microbiota transforms into a restricted flora dominated by the Firmicutes family […] to […] the microbiota transforms into a restricted flora dominated by families belonging to the phylum Firmicutes. Line 37 and through the rest of the manuscript: write all Latin names in italics.

Introduction:

The introduction effectively highlights the evolving understanding of the lung microbiome, clarifying the difference between the microbiome and microbiota. In this section, the dynamic nature of the lung microbiome is presented. Described are factors that influence lung microbiota and interactions between microbes.

Some minor editorial corrections are needed. For example, at line 96, remove the question mark after ILD.

The introduction briefly introduces the concept of operational taxonomic units (OTUs), and in my opinion, this part should be elaborated so that readers who are unfamiliar with NGS can clearly understand the advances and limitations of this type of sequencing and microbiome profiling.

The aim of the study is clearly defined, stating its intention to characterize and compare the lung microbiome in three distinct interstitial lung diseases.

In conclusion, the introduction is well-structured and provides a strong foundation for the subsequent research. However, further elaboration on the mentioned aspects would enhance its overall effectiveness.

Materials and Methods:

The "Materials and Methods" section of this manuscript provides a comprehensive description of the study's design, patient selection, and the methodologies employed for data collection and analysis.

Study Design and Patient Selection: The study is well-designed as a prospective cohort study. The inclusion and exclusion criteria are clearly defined, ensuring that the patient population selected is appropriate for the research. The decision to focus on three distinct interstitial lung diseases (IPF, HP, and unclassified ILD) is logical, and the hypothesis guiding the study is clearly presented.

Some minor editorial corrections are needed. For example, line 114: write the full name of “NIV.” Lines: 105, 107, 134, 141, 145: I would recommend bolding the following words: Inclusion/exclusion criteria, Sample preparation, Bacterial DNA extraction, and Metagenomic sequencing. Lines 132-134 should be replaced and/or repeated in the section: The Institutional Review Board Statement.

Overall, the "Materials and Methods" section is comprehensive and well-organized. However, in the later part of the review, I will make remarks regarding this section.

Results:

To this section, I have the following comments:

Line 191: It is stated that you obtained information down to the species level. However, in the manuscript, there is no information about identified species.

Additionally, please describe, in the section Materials and methods, what was the mode of sequencing; how long were the reads; what was the minimal number of reads per sample; what value was the parameter Q; what was the size of the output (number of Gb).

Lines 209-210: Must be translated into English.

Line 220-221: I recommend changing the word “pathogens” to “microorganisms.”

Discussion:

The "Discussion" section of the manuscript provides a comprehensive overview and interpretation of the study's findings, and it effectively synthesizes the study's findings and locates them in the context of existing research.

Line 339: Are the values 103 and 105 correct? Or did you mean 10 to the power of 3 and 5 (10^3, 10^5)?

References must be presented in one style; for example, see positions 1, 2, and 31 – three different styles of citation.

Summary of the Review:

The manuscript is well-structured and presents the research in a clear and organized manner. Providing more detail on the sequencing and analysis process would be beneficial for readers who want to replicate the study. The presented study—the role of the lung microbiota in ILDs—addresses an important and clinically relevant topic. The discussion section openly presents study limitations and discusses the results with literature.

Author Response

First and foremost, thank you very much for your interest in our work, for your time and support. Suggestions are always welcome as they can help us improve the paper. 

The introduction effectively highlights the evolving understanding of the lung microbiome, clarifying the difference between the microbiome and microbiota. In this section, the dynamic nature of the lung microbiome is presented. Described are factors that influence lung microbiota and interactions between microbes.

Some minor editorial corrections are needed. For example, at line 96, remove the question mark after ILD.

The introduction briefly introduces the concept of operational taxonomic units (OTUs), and in my opinion, this part should be elaborated so that readers who are unfamiliar with NGS can clearly understand the advances and limitations of this type of sequencing and microbiome profiling.

The aim of the study is clearly defined, stating its intention to characterize and compare the lung microbiome in three distinct interstitial lung diseases.

In conclusion, the introduction is well-structured and provides a strong foundation for the subsequent research. However, further elaboration on the mentioned aspects would enhance its overall effectiveness.

Materials and Methods:

The "Materials and Methods" section of this manuscript provides a comprehensive description of the study's design, patient selection, and the methodologies employed for data collection and analysis.

Study Design and Patient Selection: The study is well-designed as a prospective cohort study. The inclusion and exclusion criteria are clearly defined, ensuring that the patient population selected is appropriate for the research. The decision to focus on three distinct interstitial lung diseases (IPF, HP, and unclassified ILD) is logical, and the hypothesis guiding the study is clearly presented.

Some minor editorial corrections are needed. For example, line 114: write the full name of “NIV.” Lines: 105, 107, 134, 141, 145: I would recommend bolding the following words: Inclusion/exclusion criteria, Sample preparation, Bacterial DNA extraction, and Metagenomic sequencing. Lines 132-134 should be replaced and/or repeated in the section: The Institutional Review Board Statement.

Overall, the "Materials and Methods" section is comprehensive and well-organized. However, in the later part of the review, I will make remarks regarding this section.

Response:

We added the following fragment (depicted in red)  in the “Material and Methods” section:

The obtained template was loaded on an Ion 318 chip (ThermoFisher Scientific) and sequenced on the Ion Torrent Personal Genome Machine (ThermoFisher Scientific) with the help of the Ion PGM Hi-Q View Sequencing kit (ThermoFisher Scientific) and in single read mode. The output of the sequencing runs showed a mean reads length of 225bp, with 277470 to 1113955 reads per sample, Q20 reads between 4867974 and 231062125 bases and an output of up to 1.3Gb of data.

Results:

To this section, I have the following comments:

Line 191: It is stated that you obtained information down to the species level. However, in the manuscript, there is no information about identified species.--> see

Additionally, please describe, in the section Materials and methods, what was the mode of sequencing; how long were the reads; what was the minimal number of reads per sample; what value was the parameter Q; what was the size of the output (number of Gb).

Lines 209-210: Must be translated into English.

Line 220-221: I recommend changing the word “pathogens” to “microorganisms.”

Discussion:

The "Discussion" section of the manuscript provides a comprehensive overview and interpretation of the study's findings, and it effectively synthesizes the study's findings and locates them in the context of existing research.

Line 339: Are the values 103 and 105 correct? Or did you mean 10 to the power of 3 and 5 (10^3, 10^5)?

References must be presented in one style; for example, see positions 1, 2, and 31 – three different styles of citation.

Summary of the Review:

The manuscript is well-structured and presents the research in a clear and organized manner. Providing more detail on the sequencing and analysis process would be beneficial for readers who want to replicate the study. The presented study—the role of the lung microbiota in ILDs—addresses an important and clinically relevant topic. The discussion section openly presents study limitations and discusses the results with literature.

The rest of the changes were made in the text.